# Full-color laser displays based on organic printed microlaser arrays

Jinyang Zhao[1,2], Yongli Yan [1], Zhenhua Gao[1,2], Yuxiang Du[1,2], Haiyun Dong[1], Jiannian Yao[1,2] & Yong Sheng Zhao [1,2]

Laser displays, which exploit characteristic advantages of lasers, represent a promising next-generation display technology based on the ultimate visual experience they provide. However, the inability to obtain pixelated laser arrays as self-emissive full-color panels hinders the application of laser displays in the flat-panel sector. Due to their excellent optoelectronic properties and processability, organic materials have great potential for the production of periodically patterned multi-color microlaser arrays. Here, we demonstrate for the first time full-color laser displays on precisely patterned organic red-green-blue (RGB) microlaser matrices through inkjet printing. Individual RGB laser pixels are realized by doping respective luminescent dyes into the ink materials, resulting in a wide achievable color gamut 45% larger than the standard RGB space. Using as-prepared microlaser arrays as full-color panels, we achieve dynamic laser displays for video playing through consecutive beam scanning. These results represent a major step towards full-color laser displays with outstanding color expression.

[1] Key Laboratory of Photochemistry, Institute of Chemistry, Chinese Academy of Sciences, 100190 Beijing, China. [2] University of Chinese Academy of Sciences, 100049 Beijing, China. Correspondence and requests for materials should be addressed to Y.Y. (email: ylyan@iccas.ac.cn) or to Y.S.Z. (email: yszhao@iccas.ac.cn)

Laser displays benefit from the inherently high monochromaticity and brightness of laser emissions and hold promise for revolutionizing the display industry because of their wider achievable color gamut, higher contrast ratio, and more vivid colors than those of traditional display technologies based on incoherent broadband light sources[1–4]. The lack of appropriate self-emissive laser display panels has impeded the actualization of full-color laser displays in portable devices such as PDAs, cell phones and laptops. Therefore, developing a pixelated structure capable of multicolor lasing with an extremely wide tuning range is of great importance[5–7]. An effective strategy is to integrate discrete red, green, and blue microlasers into periodic arrays to construct display panels, where each set of RGB microlasers forms a pixel[8,9]. However, realizing such a device with traditional semiconductors has been challenging because of the intrinsic difficulties in the growth and patterning of individual semiconductor materials onto an identical substrate as a result of poor material compatibility[10,11].

Due to their excellent compatibility, large optical cross-sections and broad spectral coverage[12–16], organic materials are promising candidates for the development of full-color lasers[17–21]. The outstanding flexibility and processability of organic materials[22–26] permits their fabrication into microlaser arrays through photolithography[27], electron beam lithography[28], laser direct writing[29], and so forth[30]. However, the acquisition of full-color microlaser arrays has been hindered by the difficulties in precisely patterning individual RGB microlasers onto an identical pixelated display panel. Inkjet printing, which is a powerful tool for the mass production of multifunctional microstructures in organic flexible electronics[31–36], shows considerable potential for the large-scale fabrication and integration of photonic devices[37,38]. Therefore, developing a new method of printing that provides precise deposition of multiple inks to form regularly shaped microstructures as high-quality resonators would represent a step toward the acquisition of full-color panels for laser displays.

Herein, we demonstrate full-color laser displays with pixelated microlaser arrays as display panels composed of periodically patterned organic RGB spherical cap-shaped microcavities. The panels were fabricated by precisely positioning the microspherial cap lasers via an ultrasonic vibration-assisted inkjet printing. Red, green, and blue-emissive microlasers were obtained by doping the corresponding luminescent dyes into different spherical caps. Appropriately exciting each of the three adjacent spherical caps enabled full-color tunable lasing to build an individual RGB display pixel. On this basis, we produced full-color laser displays with the RGB microlaser pixel arrays. Moreover, using a technique of programmable laser beam scanning, we have achieved dynamic laser displays, thereby validating the applicability of as-printed panels in showing still images and videos. The outstanding performance and feasible fabrication of the organic printed microlaser arrays as display panels will support the innovation of concepts and device architectures in display technologies.

## Results

**Fabrication of organic printed RGB microlaser pixel arrays.**
Organic microlaser arrays were prepared by selectively printing organic ink solution droplets at specific positions on substrates according to the predesigned digital patterns. First, a thin layer of 1H,1H,2H,2H-perfluorooctyltriethoxysilane was coated on the substrate surface to induce a hydrophobic effect (Supplementary Fig. 1), which is the main drive to the formation of the spherical cap morphology of each droplet[39]. As schematically shown in Fig. 1a, the fabrication processes include imbibing an ink solution with a glass needle via capillary action, followed by spraying of a drop assisted by ultrasonic vibration onto a hydrophobic substrate. The ink droplets dispersed on the hydrophobic substrate exhibit a geometry of spherical cap (Supplementary Fig. 2), and become solid as the evaporation of water (Supplementary Fig. 3).

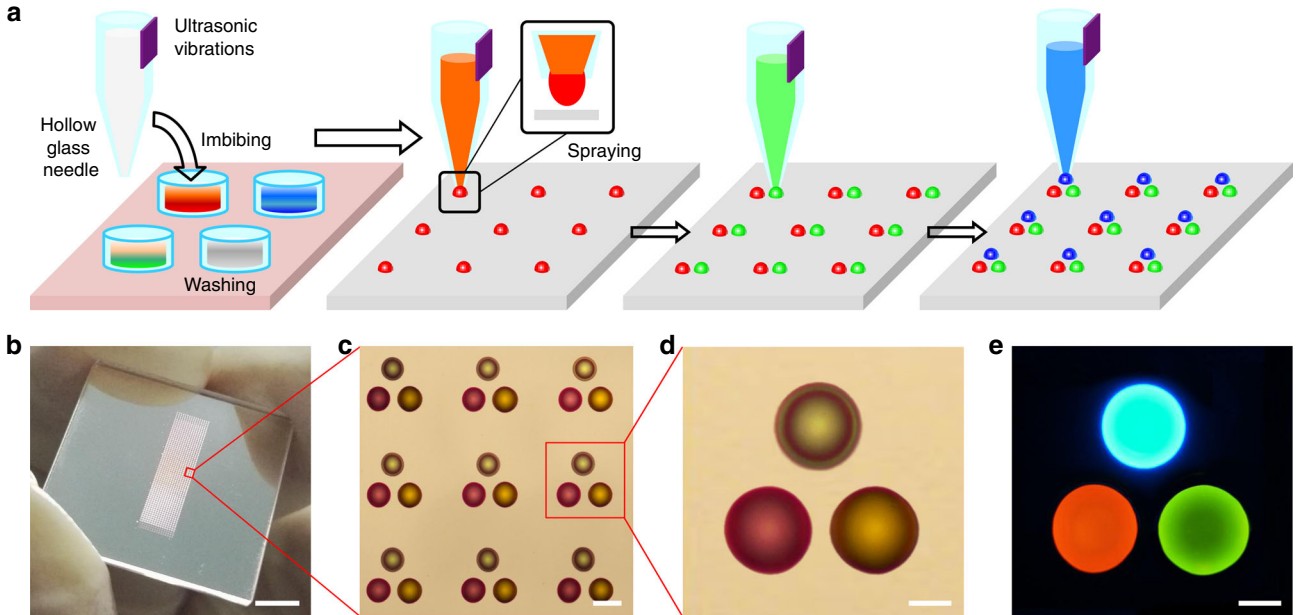

**Fig. 1** Design and fabrication of organic printed RGB microlaser pixel arrays. **a** Schematic illustration of the fabrication of organic RGB microlaser pixel arrays by ultrasonic vibration-assisted inkjet printing. The process begins with the filling of a hollow glass needle with the organic ink solution via capillary action. The solution is subsequently ejected from the glass needle under the assistance of ultrasonic vibration and then dropped onto a hydrophobic substrate. After printing the red-emissive material, green- and blue-emissive materials were printed in the same way with precise alignment. **b** Image of large-area ordered optical structures. The scale bar is 5 mm. **c, d** Microscopy images of the printed RGB pixel array showing uniform size and a well-defined pattern. The scale bars are 50 and 20 μm, respectively. **e** Fluorescence microscopy image of the printed RGB pixel under UV light radiation (330–380 nm). The scale bar is 20 μm

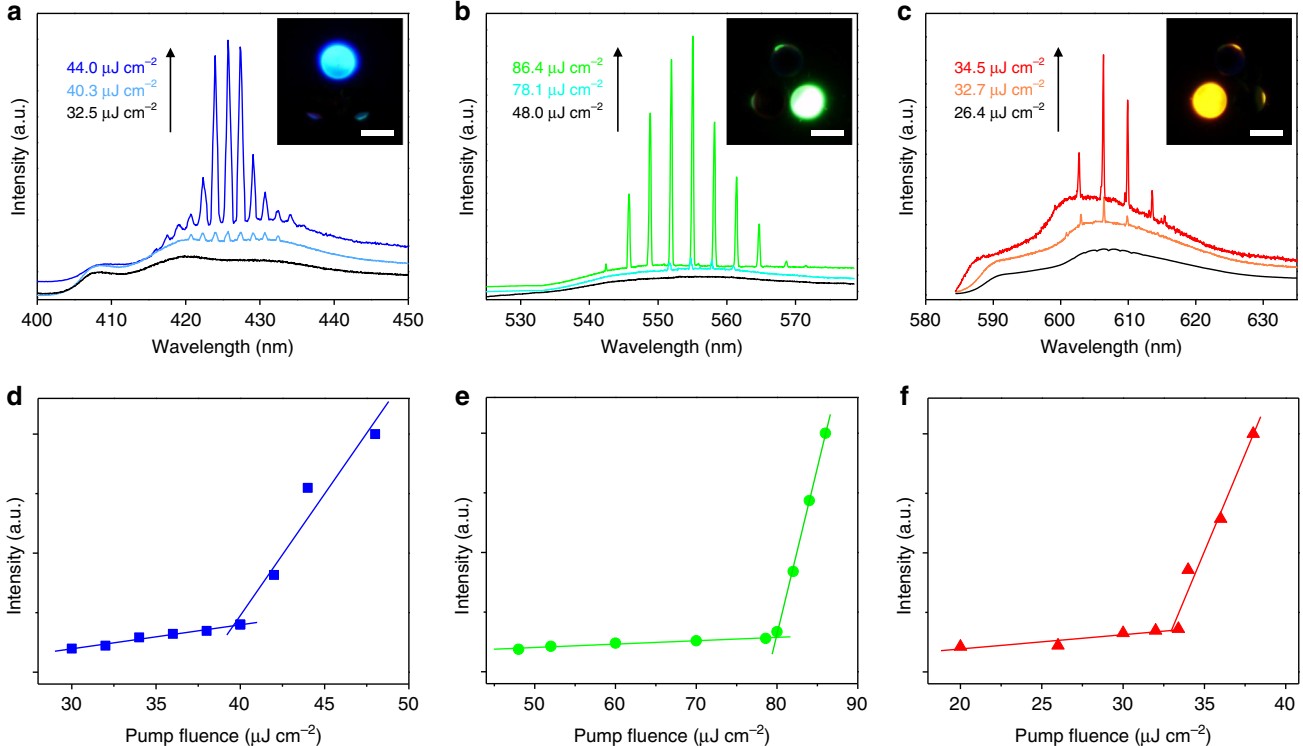

**Fig. 2** RGB lasing of an individual pixel. **a–c** Photoluminescence spectra from an individual pixel under different pump fluences with excitation on the blue-emissive spherical cap (**a**), green-emissive spherical cap (**b**), and red-emissive spherical cap (**c**), respectively. The emergence of sharp peaks indicates the occurrence of lasing behavior. Insets: corresponding microscopy images of the RGB lasing spherical caps. All scale bars are 20 μm. **d–f** Plots of the photoluminescence peak intensity at 425, 555, and 606 nm vs pump fluence, showing the lasing thresholds of 39.6, 80.0, and 33.4 μJ cm$^{-2}$, respectively

This process can be applied to most organic ink solutions (Supplementary Fig. 4) on a variety of hydrophobic substrates (Supplementary Fig. 5). The atomic force microscopy (AFM) images presented in Supplementary Fig. 6 verify that the as-prepared microstructures have well-defined spherical cap morphology with smooth surfaces. This morphology minimizes undesirable optical scattering and confines the guided emission efficiently, which is beneficial for achieving strong microcavity effects[40]. The size can be readily controlled by using the glass needle with different tip diameters (Supplementary Fig. 7) and/or varying the ultrasonic vibration strength (Supplementary Fig. 8) over a wide range (Supplementary Table 1). The printed microstructures are highly reproducible (with a uniform size on the same chip, Supplementary Fig. 9) and thus are ideal building blocks for producing microcavity arrays with a high pack density (Supplementary Fig. 10).

Each of these microspherical caps can serve as an excellent high-quality whispering-gallery-mode (WGM) cavity to support laser oscillations (Supplementary Fig. 11). The coverage range of the lasing wavelength can be tuned by incorporating various luminescent dyes into the ink. Three laser dyes were incorporated into the ink solutions: rhodamine B (RhB), fluorescein disodium salt (uranin), and stilbene 420 (S420), which have photoluminescence (PL) emissions in the red, green and blue wavebands, respectively (Supplementary Fig. 12). Sequentially printing the red-, green-, and blue-emissive inks using the same process with precise alignment produced pixelated RGB spherical cap arrays (Fig. 1a). The as-fabricated RGB spherical caps can be patterned over a very large area (Fig. 1b), and in a uniform geometric shape at precise positioning (Fig. 1c), which is essential for practical displays. Every three adjacent spherical caps function as an individual pixel (Fig. 1d). Under UV light irradiation, the pixel emits bright red, green and blue fluorescence (Fig. 1e), supporting

the feasibility of the printed microspherical cap arrays as full-color self-emissive laser display panels.

**Lasing from individual pixels**. When the dye-doped spherical caps were separately excited with a focused pulsed laser beam (355 nm, ~150 fs) in a custom microphotoluminescence system (Supplementary Fig. 13), bright ring-shaped patterns were observed at the outer boundaries (Fig. 2a–c, inset), indicating the formation of WGM resonances. The spectral evolutions with increasing laser power are shown in Fig. 2a–c. At a low pump energy with fluence <26.4 μJ cm$^{-2}$, all PL spectra were dominated by broad spontaneous emissions. When the excitation fluences exceeded their thresholds, strong laser emissions developed as a set of sharp peaks of blue, green, and red centered at 425, 555, and 606 nm, respectively. The linewidth of the individual modes narrowed down to ~0.3 nm, verifying the high quality of the obtained resonators. Moreover, the mode spacing gradually increased with decreasing spherical cap base diameters (Supplementary Fig. 14), providing an opportunity to achieve single-mode lasers, which are optimal light source for laser displays[2]. The plots of the corresponding PL intensity as a function of the pump fluence, as depicted in Fig. 2d–f, exhibit a clear knee behavior characteristic, further confirming the lasing action from the printed RGB emissive microspherical caps.

**Full-color tunable lasing in the printed pixel arrays**. This RGB lasing action paves the way for constructing full-color pixelated panels for laser display through additive color. Here, the color expression was evaluated through the emission spectra of an individual pixel (Fig. 3a). When the excitation beam was focused on the spherical cap doped with S420, lasing at ~425 nm occurred, and a blue spot was observed in the RGB pixel. As the

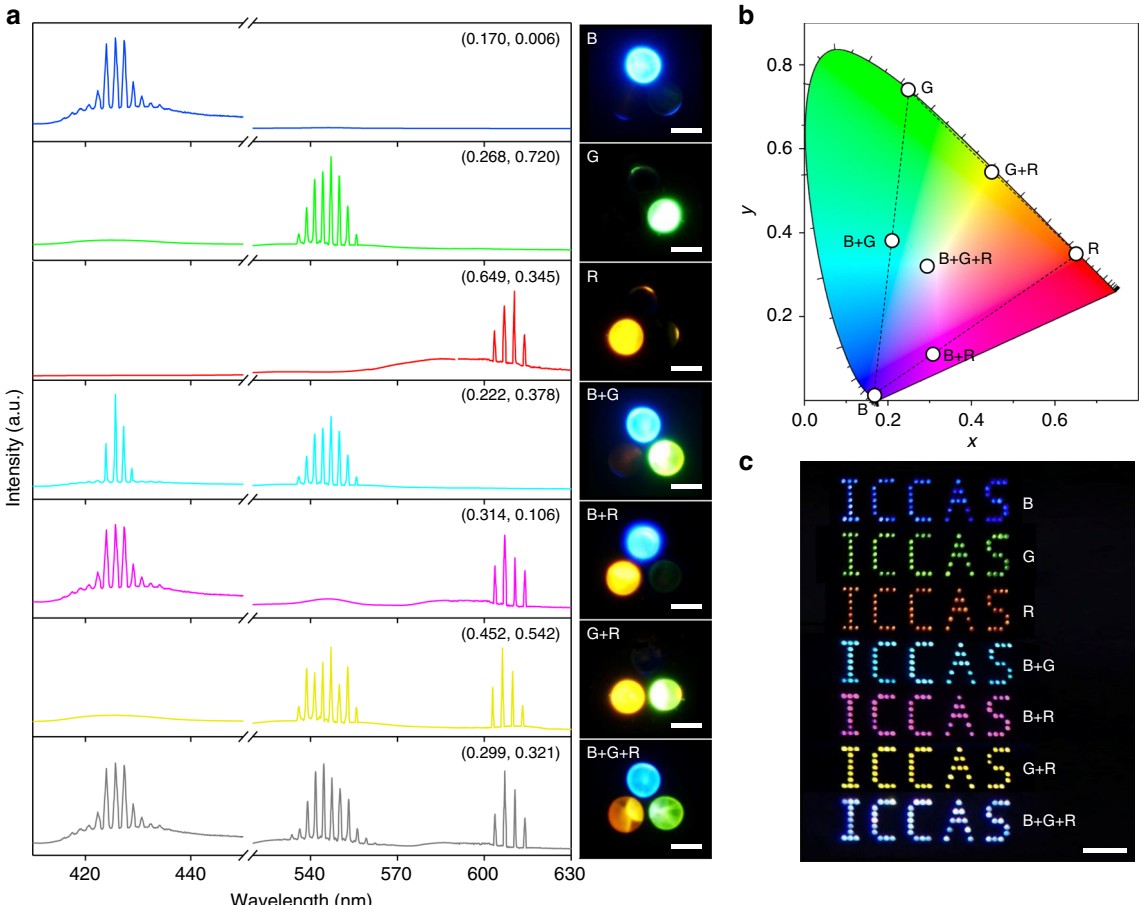

**Fig. 3** Full-color tunable lasing. **a** Lasing spectra and corresponding photoluminescence images when different positions of the RGB pixel were pumped above their thresholds. From top to bottom: the blue (B), green (G), red (R), blue and green (B + G), blue and red (B + R), green and red (G + R), and blue, green and red (B + G + R) emissive spherical caps. The numbers show the CIE1931 coordinates calculated from the corresponding spectra. All scale bars are 20 μm. **b** Chromaticity of the lasing peaks extracted from the spectra in **a**, shown as seven white circles. The dashed lines indicate the range of the achievable color gamut for the RGB pixel. **c** Far-field photograph of the "ICCAS" patterns of pixel arrays comprising different RGB emissive spherical caps. Mixed far-field emission colors of blue, green, red, cyan, magenta, yellow, and white were clearly observed from the panel under UV excitation. The scale bar is 2 mm

excitation beam was manipulated to irradiate on the spherical cap doped with uranin (or RhB), the spectra were dominated by green (or red) emission induced by the lasing from the corresponding resonators. When the blue- and green-emissive spherical caps were simultaneously pumped above their thresholds, two-color lasing (B + G) was obtained. Any light combination comprising two of the three additive primary colors in varying proportions can be generated by adjusting the manner of excitation (Supplementary Fig. 15), and simultaneous RGB lasing was realized from the pixel when all three spherical caps were integrally pumped.

Figure 3b shows the calculated chromaticity for these lasing spectra on a CIE1931 color diagram (blue, green, red, cyan, magenta, yellow, and white). The chromaticity of the balanced white lasing (B + G + R) is very close to that of the CIE standard white illuminant D65[41]. In addition, according to Grassmann's law, all colors inside the triangle defined by the three primary colors (blue, green and red) can be produced by mixing the three primary colors in appropriate proportions[41]. The printed RGB pixel can cover 45% more perceptible colors than the standard RGB space after converting to a perceptually uniform color space (Supplementary Fig. 16, Supplementary Tables 2, 3 and Supplementary Note 1). Such a large color gamut suggests superiority of the printed RGB

microlaser arrays in the production of vivid displays with excellent color saturation.

Accurate color rendering in the far-field, which is an essential issue in the design of laser displays[3,42], was further examined by taking real color photographs of pixelated laser arrays under laser irradiation. The colors were created by the mixing of light emitted from the spherical cap microlasers. We fabricated a series of microlaser arrays in an "ICCAS" pattern by selectively printing distinct inks on an identical substrate. With a built-in digital camera in a cell phone, we obtained the far-field image of the fabricated patterns, which can also be easily captured by naked eyes. It is shown in Fig. 3c that blue, green and red "ICCAS" patterns appeared when the arrays were fabricated solely with inks doped with S420, uranin and RhB, respectively. Moreover, cyan, magenta, and yellow mixed lasing emissions were observed from the corresponding patterns comprising R + G, B + R, and G + R emissive spherical caps, respectively. Pumping the RGB pixelated "ICCAS" pattern resulted in simultaneous mixing of the lasing emissions from all three spherical caps, producing a white color. The excellent implementation of the multi-primary spatial-temporal color synthesis indicates that the printed pixelated RGB laser arrays are good candidates for use in full-color laser display panels.

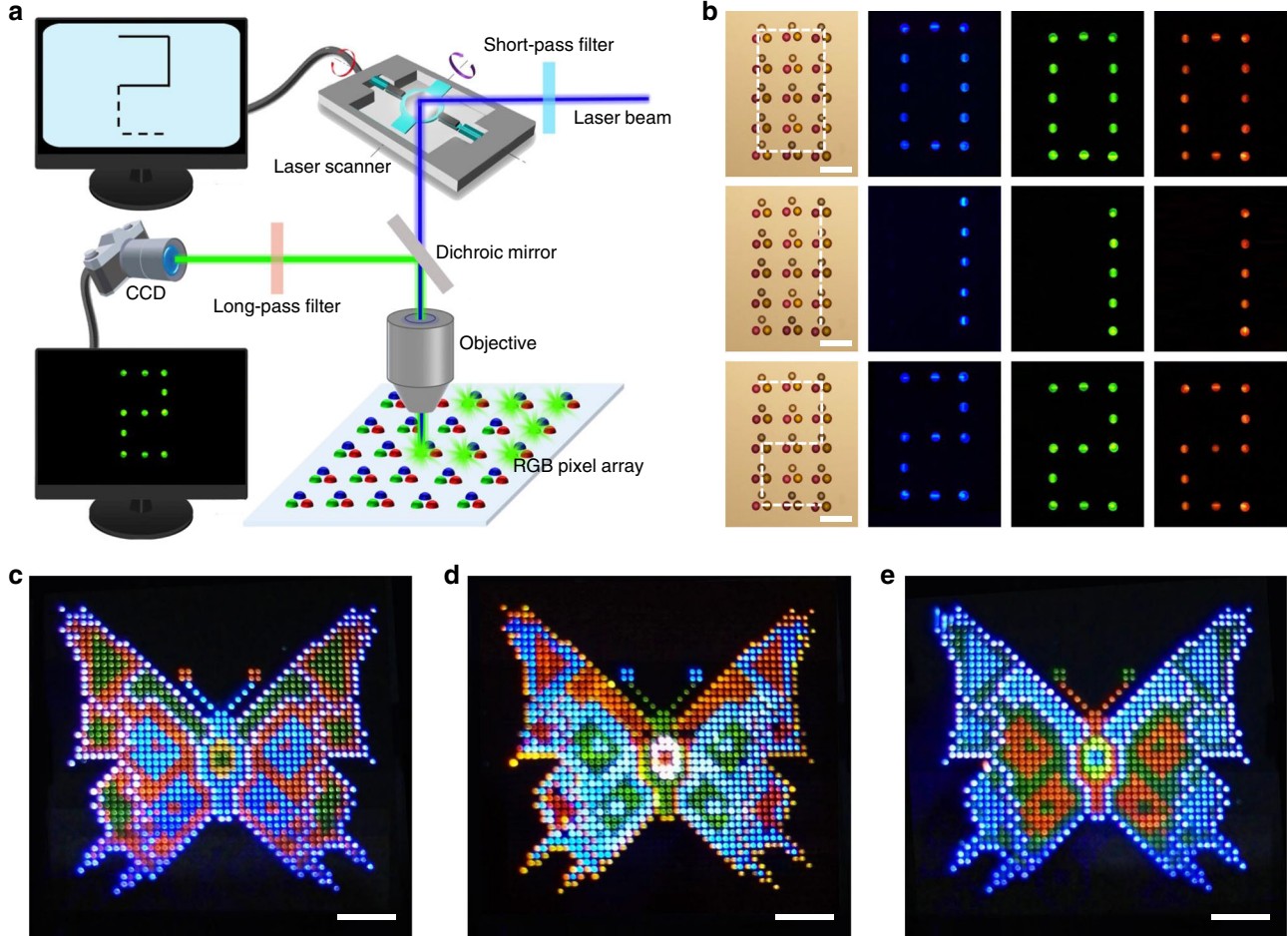

**Fig. 4** Full-color laser display. **a** Concept for full-color laser display based on programmable excitation. Briefly, a 355 nm pulsed laser is directed into a fast three-dimensional laser scanner, which is controlled so that the laser beam scans the RGB laser pixel array along pre-defined paths. As a result, tunable color images can be achieved through spatially varied optical pumping. **b** Microscopy images (left column) and photoluminescence images (right three columns) of a RGB pixel array, showing the ability to display multi-color Arabic numbers using the as-developed display panel. The blue, green and red 0, 1, or 2 can be displayed by exciting the corresponding spherical caps along the laser scanning path, that is, the white lines in the left column of the panel in Fig. 4b. All scale bars are 200 μm. **c–e** Images of 2.2-cm (diagonal) laser display prototypes using an identical full-color panel with a 44 × 44 pixel array. All scale bars are 3 mm

**Full-color laser display with the printed panels**. Laser displays can be realized on the printed RGB pixel arrays by controllably lighting up pixels in different locations[43] as any image can be viewed as a matrix of pixels. Combinations of different lasing pixels with distinct colors were generated by sequentially exciting the pixels by fast scanning of the excitation beam along defined paths on the RGB pixel array (Fig. 4a). Taking advantage of the human eye's persistence of vision, we were able to observe a complete color image when the scanning assignment was fulfilled within the permissible time (~0.02 s). Illustrated in Fig. 4b presents a 3 × 5 pixel array displaying "0", "1" and "2" numbers by selectively exciting spherical caps along the scanning paths depicted as the white lines in the left column. Arbitrary character strings can be displayed on a pixelated panel (Supplementary Fig. 17) by setting up an appropriate scan pathway, thereby validating the displays of the desired text messages.

The display of colorful images can be realized through programmable excitation control. Various images were translated into their corresponding scanning pathways, and beam scanning was performed by rapidly rotating a galvanometer mirror to generate complete pictures through the lasing of specific pixels along the path. Consecutively modulating the predesigned scanning paths one after another would induce a switching of

the displayed patterns for showing still images or scrolling on-screen information. As an example, Fig. 4c–e present the alternate displays of three butterfly images with different color distributions on an identical panel (2.2 cm in diagonal size, 44 × 44 pixels); the images demonstrate a method of realizing full-color dynamic laser displays. Moreover, further accelerating the scanning speed until the intervals between two proximate frames are not perceivable, would result in the illusion of moving images (see the Supplementary Movie 1), indicating that the accurately positioned RGB microlaser arrays can be well applied in the display of animations and movies with excellent color expression. These prototypes manifest the applicability of the printed microlaser arrays using various organic functional materials, for self-emissive full-color laser displays.

## Discussion

We have demonstrated full-color laser displays with self-emissive organic pixelated microlasers constructed using a technique of vibration-assisted inkjet printing. Programmable patterns of red, green, and blue spherical cap-shaped microlaser arrays were produced through sequential printing of the corresponding luminescent inks with precise positioning, in which each set of

RGB microlasers constituted an individual display pixel. A single RGB pixel was tuned to lase over a wide range of visible color via controlled pumping, thereby facilitating accurate color rendering in the far-field. On this basis, a prototype of full-color laser displays was realized using the pixelated microlaser arrays as display panels. Dynamic laser displays were demonstrated by using an identical panel with programmable excitation control; these displays are appropriated for showing still images and videos. These results bring us one step closer to developing high-performance, easy-to-fabricate, large-area flat-panel laser displays, and laser illumination devices.

## Methods

**Ink materials**. Bovine serum albumin (BSA, 98%), Araldite 506 epoxy resin and NOA86, which were purchased from J&K Scientific Ltd. (Beijing, China), Sigma-Aldrich (Shanghai, China), and Lienhe Fiber Optic Supplies (Beijing, China), respectively, were selected as the matrix material to create the high-quality resonators due to their outstanding flexibility. Glycerin (>99.5%) purchased from Alfa Aesar (Shanghai, China) was used to adjust the viscosity of the BSA aqueous solution, which is a key factor in preventing shrinkage of the printed structures after solvent evaporation. The materials were used as received without further treatment, unless stated otherwise.

**Laser dyes**. Stilbene 420 (S420, 97%), fluorescein disodium salt (uranin, 99%), and rhodamine B (RhB, 99%), which exhibit photoluminescence at blue, green, and red wavebands, respectively, were selected as gain media for lasing. The three laser dyes were purchased from J&K Scientific Ltd. (Beijing, China), Acros Organics (Beijing, China), and TCI (Shanghai, China), respectively. The dyes were purified by crystallization in water.

**Other materials**. 1H, 1H, 2H, 2H-Perfluorooctyltriethoxysilane (PTES, 97%), which was purchased from Sigma-Aldrich (Shanghai, China), was used to modify the surface of the substrate to induce a hydrophobic effect.

**Substrate preparation**. The substrates, including the glass, silver mirror and silicon wafer, were cleaned in oxygen plasma for 5 min and subsequently modified with PTES by evaporation at 100 °C for 30 min in an evacuated flask to enhance the hydrophobic effect, which is essential for the formation of microcavities. In this work, silver mirrors were chosen as the substrates for display panels because of their high reflectivity (99.5%) in the visible range, which is beneficial for achieving low-threshold laser emission. A thin layer of magnesium fluoride (100 nm) was coated on the silver mirrors to further suppress energy leakage into the substrates.

**Fabrication of spherical cap shaped microlasers**. The spherical caps were obtained by depositing inks on the hydrophobic substrates using a GIX$^{TM}$ Microplotter$^{TM}$ II from Sonoplot INC. The inks were prepared by dissolving 2000 mg of BSA in a mixed solution of 5 mL of deionized water and 2 mL of glycerin. The printing process included imbibing ink solution with a glass needle via capillary action and spraying a drop assisted by ultrasonic vibration. Due to the hydrophobic effect, the droplet self-assembled into a spherical cap-shaped structure with a circular shape. After drying in air for one day, solid-state spherical cap microstructures were obtained.

**Fabrication of pixelated microlaser arrays**. The spherical cap laser arrays were fabricated by sequentially depositing luminescent inks at specific positions onto substrates according to predesigned digital patterns. The red-, green- and blue-emissive inks were prepared by dissolving 20 mg each of RhB, uranin, and S420 in 5 mL of BSA aqueous solution (400 mg L$^{-1}$), followed by the addition of 2 mL of glycerin. The three luminescent inks were printed in the same manner with precise alignment to form periodic RGB pixel arrays. Before printing, the glass needle was cleaned by washing (deionized water) to avoid ink contamination.

**Characterizations of individual microlasers**. The morphology of the as-prepared microspherical caps was examined by atomic force microscopy (AFM, NTEGRA Solaris, NT-MDT). Bright-field and fluorescence microscopy images were acquired using inverted fluorescence microscope (Nikon, Ti-U) under the excitation of a mercury lamp (330–380 nm). The optically pumped lasing measurements for individual pixels were conducted on a custom microphotoluminescence system. The excitation pulses (355 nm) were generated from an optical parametric amplifier (Light Conversion TOPAS) pumped by a regenerative amplifier (Spectra Physics, 800 nm, 150 fs, 1 kHz), which was in turn seeded by a mode-locked Ti: sapphire laser (Mai Tai, Spectra Physics, 800 nm, 150 fs, 80 MHz). The excitation laser was filtered with a 400 nm short-pass filter and then focused down to a 20 μm diameter spot through an objective lens (Nikon CFLU Plan, ×20, N.A. = 0.5) as a nearly uniform pump source. The power at the input was altered by a neutral

density filter. The emissions from the individual spherical caps were collected by the same objective with a back-scattering configuration and analyzed by the spectrometer after removing the excitation beam with a 400-nm long-pass filter. A far-field photograph was taken from the panels irradiated by 355-nm laser pulses using a build-in digital camera in a cell phone; the excitation beam was expanded to evenly illuminate the field of view.

**Laser displays through beam scanning**. Dynamic displays on the printed RGB microlaser panels were performed by controllably lighting up pixels in different locations. The output laser beam from a 355-nm pulsed laser was aligned through a 400-nm short-pass filter and directed into a three-dimensional laser scanner, and refocused onto the display panels. The focused laser beam was controlled by rotating the galvanometer mirror, and the scanning pathway across the panels was precisely controlled by laser scanning software according to a preprogrammed pattern. A fast imaging CCD was employed to record the lasing emissions from the pixels while the excitation beam was successively modulated, resulting in observable images and videos.

**Simulation**. The electric field intensity distributions were obtained using the finite-difference time-domain (TDFD) method and the TDFD source code can be accessed from http://www.fdtdxx.com.

## Data availability

The data that support the findings of this study are available from the corresponding author upon reasonable request.

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

## Acknowledgements

This work was supported financially by the Ministry of Science and Technology of China (Grant No. 2017YFA0204502), the National Natural Science Foundation of China (Grant Nos. 21773265, 21790364, and 21533013), and the Chinese Academy of Sciences (XDB12020300).

## Author contributions

Y.S.Z. conceived the original concept. Y.S.Z and Y.Y. supervised the project. J.Z. and Y.Y. designed and performed the experiments and prepared the materials. J.Z., Y.D., H.D., and Z.G. performed the optical measurements. J.Z., Y.Y., J.Y., and Y.S.Z. analyzed the data and wrote the paper. All authors discussed the results and commented on the manuscript.

## Additional information

**Competing interests:** The authors declare no competing interests.

