## [Peer Review File · Nature Communications]

Reviewers' comments:

Reviewer #1 (Remarks to the Author):

Authors reported in this manuscript an innovative method of creating micro-scale laser cavities at individual pixel level, for the purpose of color display. The fabrication of the micro-laser cavities was accomplished by dispersing dye-containing aqueous ink on a hydrophobic surface through a sharp glass needle assisted by ultrasonic vibration. The spontaneous formation of micro-scale ink droplets following air drying for 24 hours creates solid-state hemispherical structure of tens of micrometer in diameter. The micro-hemispheres support the whispering gallery mode (WGM) resonance, which can be optically pumped at UV wavelength using femtosecond laser at UV in reaching the lasing condition. The center wavelength of the collective lasing mode from the micro-hemispherical structures can be readily tuned by changing the selection of dye to be incorporated into the ink. Thus, patterning microhemispheres comprising distinct Red, Green, and Blue lasing modes in a periodic array makes it possible for the experimental demonstration of full-color laser displays. The use of micro-laser cavities as the discrete pixel not only significantly improve its brightness, but also allows for precise control of color expression. Authors demonstrated in this work a wide achievable color gamut 45% larger than the standard RGB space. Clearly the demonstration of micro-laser array as the color display panel is a novel contribution of this manuscript. However, there any detailed question in regards to the intend applications for color display:

1. Authors stated in page 3, "The lack of appropriate self-emissive display panels has impeded the actualization of full-color laser displays in portable devices such as PDAs, cell phones and laptops." To reviewer's knowledge, the self-emissive display panel does exist, which is the organic light-emitting diode (OLED). The OLED panel has now been widely used in the consumer electronic market, ranging from cell phone, digital camera, and ultrathin flat-panel TV. Authors may want to elaborate the performance of their micro-laser display with respect to the state-of-art OLED technologies, especially the achievable color gamut.
2. Authors defined the fabricated pixel as "hemisphere". However, as shown in the Supplementary Figure 3, the resulting structure is ~ 4 μm in height and ~ 17 μm in diameter, which is far from being a hemisphere. Additionally, the measured cross-sectional profile exhibits appreciable asymmetry from left to right. It would be beneficial to the reader to compare the measured cross-sectional profile against the idea spherical profile. Furthermore, it seems authors have performed extensive study of the later dimension of the hemisphere, but with very limited attention to its morphology in 3D. The later would in fact be more crucial for the formation of the lasing modes.
3. As the most essential component of the reported work, authors may want to elaborate more on the fabrication methods.
 - a. What's the contact angle upon hydrophobic coating?
 - b. What's the 3D geometry of the ink droplet upon dispersion?
 - c. Authors only mentioned the droplet being dried in air for a day. What's the underlying drying mechanism? Is it the gelation of the BSA solution? What's the final composition of the "dried" hemisphere.
 - d. What's the volume shrinkage upon drying?
 - e. What's the highest packing density can be realized using the reported fabrication method.
4. In case of the dried hemisphere has high BSA content, what's its lifetime given a realistic emission power for general display application. What's the possibility failure modes due to melting, oxidation of the BSA matrix, and photobleaching of the organic dye molecules? Will the material be robust enough to withstand the high pumping power in supporting the suggest single-mode lasing mode?
5. Using an expensive femto-second laser may defeat the purpose for array display. It is possible due to the weak absorption of Red and green dye at UV wavelength (Supplementary Figure 8). Is it possible to pump the micro-laser pixel electrically?
6. It seems the formation of the WGM resonance is the necessary requirement to support the lasing mode at individual pixel. How does that affect the angular distribution of the corresponding

stimulated emission modes?

The reported idea is clearly novel and represents an important step towards full-color laser display. However, the lack of scientific rigor in the reported work makes it difficult to fully assess its impact on the potential application for full-color display.

Reviewer #2 (Remarks to the Author):

The manuscript submitted by YS Zhao et al. reports the realization of a display scheme based on a multicolour organic laser array. Each pixel in the array is composed of three individual hemisphere WGM lasers with RGB primary colours, therefore both the colour and the brightness of each pixel can be precisely tuned. The advantage of their approach lies in the simple fabrication in which the laser location and hemisphere droplet size of the lasers can be well determined. Meanwhile, the laser cavity is naturally formed by the hemisphere configuration, thus no external cavity is needed. The related research is very significant and the results reported here are interesting. However, as implementation of laser display is a big engineering issue, the reported demonstration is just an early step towards practical application. Nevertheless, the authors have indeed shown innovation in concept and mechanism in the field of display. In my view, the manuscript may be accepted, subject to the proper addressing of the following comments.

1. The authors have demonstrated both static and dynamic displays, but the obtained images and videos were obtained by a CCD camera. Therefore, the final display performance is still limited by the CCD camera. Then what's the advantage of this approach?

2. As pointed out in this manuscript, for laser display single mode lasers are preferred. Actually, a single mode hemisphere laser has been reported before [Sci. Rep. 2, 244(2012)]. However, the demonstrated images were obtained by multimode lasers, which actually missed the advantage of laser display. I wonder whether single mode laser display is possible in this approach, considering the individual pixel excitation.

3. The final target of laser display will be on either flat panel display or projector display. For projector display, the time duration of lasing should be considered. As current approach is using a fs laser for excitation, I wonder how long is the time duration. If the lasing duration is too short, then a CW excitation is necessary.

4. For flat panel display, electrical driven devices are necessary, Can the authors comment on the feasibility of electrical pumping?

Responses to Reviewer 1

Comment: *Authors reported in this manuscript an innovative method of creating micro-scale laser cavities at individual pixel level, for the purpose of color display. The fabrication of the micro-laser cavities was accomplished by dispersing dye-containing aqueous ink on a hydrophobic surface through a sharp glass needle assisted by ultrasonic vibration. The spontaneous formation of micro-scale ink droplets following air drying for 24 hours creates solid-state hemispherical structure of tens of micrometer in diameter. The micro-hemispheres support the whispering gallery mode (WGM) resonance, which can be optically pumped at UV wavelength using femtosecond laser at UV in reaching the lasing condition. The center wavelength of the collective lasing mode from the micro-hemispherical structures can be readily tuned by changing the selection of dye to be incorporated into the ink. Thus, patterning microhemispheres comprising distinct Red, Green, and Blue lasing modes in a periodic array makes it possible for the experimental demonstration of full-color laser displays. The use of micro-laser cavities as the discrete pixel not only significantly improve its brightness, but also allows for precise control of color expression. Authors demonstrated in this work a wide achievable color gamut 45% larger than the standard RGB space. Clearly the demonstration of micro-laser array as the color display panel is a novel contribution of this manuscript. However, there any detailed question in regards to the intend applications for color display.*

Response: We thank the reviewer for the appreciation of our work. We are also very grateful for the insightful comments and suggestions, which help us to be more precise about the presentation and interpretation of the scientific rigor and importance of our work. In the following, we provide concrete responses to the comments and suggestions point-by-point.

Question 1: *Authors stated in page 3, “The lack of appropriate self-emissive display panels has impeded the actualization of full-color laser displays in portable devices such as PDAs, cell phones and laptops.” To reviewer’s knowledge, the self-emissive display panel does exist, which is the organic light-emitting diode (OLED). The OLED panel has now been widely used in the consumer electronic market, ranging from cell phone, digital camera, and ultrathin flat-panel TV. Authors may want to elaborate the performance of their micro-laser display with respect to the state-of-art OLED technologies, especially the achievable color gamut.*

Response 1: We appreciate for the reviewer's comment. As the reviewer pointed, OLED panels are one kind of self-emissive display panels, which have been widely used in the consumer electronic market, ranging from cell phone, digital camera and ultrathin flat-panel TV. In comparison with the traditional display technologies based on broadband light sources, laser displays can produce wider achievable color gamut, higher contrast ratio, and more vivid colors due to the usage of spectrally pure lasers. Here, we want to elaborate the performance of micro-laser displays with respect to the state-of-art OLED technologies. Therefore, we modified the statement as *"The lack of appropriate self-emissive laser display panels has impeded the actualization of full-color laser displays in portable devices such as PDAs, cell phones and laptops"* in Page 3 to avoid possible misunderstanding.

Question 2: *Authors defined the fabricated pixel as "hemisphere". However, as shown in the Supplementary Figure 3, the resulting structure is ~ 4 μm in height and ~ 17 μm in diameter, which is far from being a hemisphere. Additionally, the measured cross-sectional profile exhibits appreciable asymmetry from left to right. It would be beneficial to the reader to compare the measured cross-sectional profile against the idea spherical profile. Furthermore, it seems authors have performed extensive study of the later dimension of the hemisphere, but with very limited attention to its morphology in 3D. The later would in fact be more crucial for the formation of the lasing modes.*

Response 2: Thanks a lot for the reviewer's comments. As the reviewer concerned, the height of obtained microstructure is much smaller than the base radius, which is far from being a hemisphere. Strictly speaking, the printed microstructure is more like a spherical cap, which can be regarded as a portion of a sphere cut off by a plane. In geometry, only if the plane passes through the center of the sphere, so that the height of the cap is equal to the radius of the sphere, the spherical cap is called a hemisphere. Through such spherical caps have be called hemispheres extensively in the field of materials (*Adv. Mater.* **24**, OP60-OP64 (2012); *Sci. Rep.* **2**, 244 (2012)), according to the suggestion, we named the printed structures as spherical caps in the revised manuscript for rigorousness.

Figure R1 Atomic force microscopy images of a typical printed spherical cap. **a**, 3D-AFM image, **b**, 2D-AFM image and **c**, corresponding cross-sectional profile of the printed microstructure. The scale bar is 10 μm .

The spherical caps were fabricated by depositing inks on hydrophobic substrates. Due to the hydrophobic effect, the droplet self-assembled into a spherical cap and became solid due to evaporation of water. Thus, the cross-sectional profile should be symmetric because of the isotropic self-assembly. The observed asymmetry in the Figure S3 can be ascribed to the tilted substrate. So we conducted AFM measurements again and replaced the original figure with Figure R1. It is shown that as-fabricated microstructures possess a well-defined spherical cap morphology and smooth surface, which are favorable for Whisper-Gallery-Mode (WGM) resonance.

Benefiting from the smooth surface and the circular bottom plane, the printed spherical caps function as typical WGM-type resonators, and their photonic performances depends heavily on structural parameters. We numerically simulated three-dimensional local electric field

distribution within a spherical cap with finite-difference time-domain method, which was displayed in Supplementary Figure 11. Notably, the electric field of light is almost localized within the spherical cap, indicating the photons are confined by the total reflection at the interface between the spherical cap and air. The electric field in the horizontal plane shows a distribution of periodic patterns in the azimuth direction, demonstrating a characteristic WGM resonance. The profile in vertical plane reveals that the travelled light mainly concentrates around the bottom part, implying that the height may not as critical as the lateral dimensions in our case.

Figure R2 Lasing characteristics of two individual spherical caps with different heights. **a-b**, Tilted optical microscopy images. The scale bar is 20 μm . Insets show the respective cross-sectional profiles. **c**, PL spectra of the two spherical caps.

Aim to experimentally clarify the relationship between the height and lasing performance, we fabricated two spherical caps with the same ink on substrates of varied wettability. Figure R2 a-b Show the titled optical microscopy images and corresponding cross-sectional profiles of two spherical caps. The two spherical caps possess an identical diameter of 21.45 μm , while their heights are 4.84 μm and 3.39 μm , respectively. Under irradiation of a femtosecond laser, the two spherical caps emit strong fluorescence. Once the pump energy exceeded the lasing thresholds, as shown in Figure R2c, the recorded fluorescence spectra from the two spherical caps exhibit several sharp peaks, which is a characteristic of multimode operation. There are no obvious differences between the spectra from the two printed spherical caps, verifying the little

dependence of lasing performance on the height in our case. It is worth mention that this conclusion is not true when the height is too small (less than half wavelength) to support optical resonance because of the severe leakage of light energy to substrate or air.

Question 3: *As the most essential component of the reported work, authors may want to elaborate more on the fabrication methods.*

a. What's the contact angle upon hydrophobic coating?

b. What's the 3D geometry of the ink droplet upon dispersion?

c. Authors only mentioned the droplet being dried in air for a day. What's the underlying drying mechanism? Is it the gelation of the BSA solution? What's the final composition of the "dried" hemisphere.

d. What's the volume shrinkage upon drying?

e. What's the highest packing density can be realized using the reported fabrication method.

Response 3: Thanks a lot. As the reviewer summarized, we report an innovative method of creating micro-scale laser cavities at individual pixel level, for the purpose of color laser display. Thus, the fabrication methods should be elaborated more.

a. About the contact angle of substrates.

The process of ink interacting with the substrate is critical to the formation of microcavities for lasing. The contact angle is the angle at the interface where ink, air and substrate meet, and its value is a measure of how likely the surface is to be wetted by the ink. Low contact angle values represent a tendency of the ink to spread and adhere to the surface, whereas high contact angle values show the surface's tendency to repel ink. The most common method for surface-wetting characterization is sessile-drop goniometry. This method determines the contact angle from the shape of the droplet and can be applied to a wide variety of materials. Here, the contact angle of substrate was determined to be 49.0°, and increased to 60.7° through a hydrophobic treatment (Figure R3). We have modified the manuscript in Page 5 as "*First, a thin layer of 1H, 1H, 2H, 2H-perfluorooctyltriethoxysilane was coated on the substrate surface to induce a hydrophobic effect (Supplementary Fig. 1), which is the main drive to the formation of the spherical cap morphology of each droplet.*" and added this figure in Supplementary Material as Figure 1.

Figure R3 Contact angle measurements of ink droplets on substrates before/after hydrophobic treatment.

b. About the 3D geometry of the ink droplet.

Figure R4 Tilted optical microscopy image of an ink droplet upon dispersion. The scale bar is 40 μm .

As shown in Figure R3, the ink droplet upon dispersion has a geometry of spherical cap due to hydrophobic effect. In addition, we obtained the tilted optical microscopy image of an ink droplet on a substrate (Figure R4), further confirming this morphology. Therefore, we added this figure in Supplementary as Fig. 2, and revised corresponding part in manuscript as “*The ink droplets dispersed on the hydrophobic substrate exhibit a geometry of spherical cap (Supplementary Fig. 2), and become solid as the evaporation of water (Supplementary Fig. 3)*” (Page 5).

c. About the underlying drying mechanism and final composition of the “dried” microstructures.

Figure R5 Record masses of dispersed ink droplets at different times.

In order to clarify the drying mechanism, we recorded the masses of dispersed ink droplets at different time, which are presented in Figure R3. The mass decreases as time elapses in the first 1.5 hours and stabilizes in the subsequent several hours. The initial sharp reduction in mass should be attributed the water evaporation of inks because of negligible weight loss from other two constituents of the inks (BSA and glycerin, see Method). Therefore, the underlying drying mechanism of the ink droplets is ascribed to the evaporation of water contained in the inks. The final compositions of solid spherical cap are glycerin, BSA, and small amount of water, with mass fractions of 52.38%, 33.17%, and 14.45%, respectively. Accordingly, we added the related discussion in Supplementary as Figure 3e.

d. About the volume shrinkage.

Figure R6 illustrates the volume shrinkage of an identical spherical cap before and after drying. The ink droplet dispersed on the hydrophobic substrate exhibit a geometry of spherical cap (Figure R6a), and the height and radius can be determined to be 64.32 and 108.83 μm (Figure R6a), respectively. Assuming that the dispersed ink droplet has a perfect spherical cap geometry, we calculated the volume to be $1.135 \times 10^6 \mu\text{m}^3$ according to the formula $V = \pi H^2(R - H/3)$. Shown in Figure R6c is the tilted optical microscopy image of the identical spherical cap after drying. There is an obvious reduction in height, while the base radius remains unchanged. Similarly, we obtained the geometric parameters and calculated a volume of $0.703 \times 10^6 \mu\text{m}^3$. Thus, the volume shrinkage is $(1.135 - 0.703) / 1.135 = 38\%$. Therefore, we added this figure in Supplementary as Fig.

3a-d, and revised corresponding part in manuscript as “*The ink droplets dispersed on the hydrophobic substrate exhibit a geometry of spherical cap (Supplementary Fig. 2), and become solid as the evaporation of water (Supplementary Fig. 3)*” (Page 5).

Figure R6 Volume shrinkage of a typical printed spherical cap. **a-b**, Tilted optical microscopy images of an ink droplet on a substrate without **(a)** /with **(b)** fitted geometric parameters. **c-d**, Tilted optical microscopy image of the dried droplet without **(c)** /with **(d)** fitted geometric parameters. All scale bars are 20 μm .

Figure R6 illustrates the volume shrinkage of an identical spherical cap before and after drying. The ink droplet dispersed on the hydrophobic substrate exhibit a geometry of spherical cap (Figure R6a), and the height and radius can be determined to be 64.32 and 108.83 μm (Figure R6a), respectively. Assuming that the dispersed ink droplet has a perfect spherical cap geometry, we calculated the volume to be $1.135 \times 10^6 \mu\text{m}^3$ according to the formula $V = \pi H^2(R - H/3)$. Shown in Figure R6c is the tilted optical microscopy image of the identical spherical cap after drying. There is an obvious reduction in height, while the base radius remains unchanged. Similarly, we obtained the geometric parameters and calculated a volume of $0.703 \times 10^6 \mu\text{m}^3$. Thus, the volume shrinkage is $(1.135 - 0.703) / 1.135 = 38\%$.

e. About the packing density.

Figure R7 Optical microscopy image of a closely packed microspherical cap pattern. The scale bar is 5 μm .

As the reviewer concerned, the packing density is an important indicator of the reported inkjet printing method to the fabrication of display panels. Here, the highest packing density was estimated with a closely-packed spherical cap array (Figure R7), which was prepared using a glass needle with a tip diameter of 5 μm . In geometry, close-packing of equal spherical caps is a dense arrangement of congruent spherical caps in an infinite, regular arrangement (or lattice). Each spherical cap (with a base diameter of 6.5 μm) has six neighbors, and the center-to-center spacing of adjacent spherical caps is a simple honeycomb-like tessellation with a pitch (distance between spherical caps centers, 7.1 μm). Thus, the pack density should be the fraction of space occupied by the spherical cap in the lattice, which is exhibited in the Fig. R7 as the white rhombus. The areas of individual spherical cap and rhombus are 33.2 μm^2 and 43.3 μm^2 , respectively. So, the highest packing density for spherical cap arrays printed with a 5 μm glass needle is 76.7%. We can fabricate a 140 \times 164 spherical cap arrays on a panel of 1 \times 1 mm through the reported inkjet printing method, and the total number of the microcavities is 22960. According to the suggestion, we revised the manuscript in Page 5 as “*The printed microstructures are highly reproducible (with a uniform size on the same chip, Supplementary Fig. 6) and thus are ideal building blocks for producing microcavity arrays with a high pack density (Supplementary Fig. 10)*” and added the related discussion in Supplementary Figure. 10.

Question 4: *In case of the dried hemisphere has high BSA content, what's its lifetime given a realistic emission power for general display application. What's the possibility failure modes due to melting, oxidation of the BSA matrix, and photobleaching of the organic dye molecules? Will the material be robust enough to withstand the high pumping power in supporting the suggest single-mode lasing mode?*

Response 4:

Figure R8 Normalized lasing intensity as a function of laser scanning time.

Thanks a lot for pointing out this issue, in particular with respect to failure modes. According to the suggestion, we recorded the PL intensity of a green-emissive spherical cap, whose lasing threshold is much higher than the other two dye doped microstructures. The excitation beam was fast scanned, and excitation fluence was fixed at $85 \mu\text{J}/\text{cm}^2$, which is strong enough to give a realistic emission power for general display application. As shown in Figure R8, the lasing intensity falls down slowly with scanning time. The lifetime is far behind that of current commercial display technology. However, the most important point we want to claim in this work is that the use of micro-laser cavities as the discrete pixel not only significantly improves its brightness, but also allows for precise control of color expression. We also demonstrated a wide achievable color gamut 45% larger than the standard RGB space, showing a significant advantage over other competitors. Laser displays are attracting increasing interest for the use of higher-quality displays for portable devices and hold great promise for revolutionizing the display industry. But now, the development of self-emissive laser displays is still in infancy and

need great efforts from broad research communities, including material sciences, chemistry, and physics.

Despite of excellent device characteristics, full realization of laser displays has been hampered due to the device stability. The most probable failure mode of as-fabricated display panels is that there is no lasing from individual laser pixel. The pixel's failure is usually related with its constitutional and structural changes induced by long-time high-power laser irradiation. The laser luminance might degrade or even disappear when the doped organic dyes are photobleached. The BSA components might be melted by continuous high-power laser pumping, resulting in the failure of lasing and displaying images. So, it is necessary to improve the reliability of individual laser pixels through material and structural design. Utilizing novel robust materials with improved optoelectric properties could decrease the lasing threshold remarkably and thus minimize the influence from excitation laser. We are confident that reliable laser display panels would be obtained in the near futures considering the rapid development of optoelectrical materials with high-performance.

Figure R9 Photoluminescence spectra from an individual red emissive spherical cap under different pump fluences. Inset shows the corresponding microscopy image. The scale bar is 10 μm .

As pointed, single mode lasers are preferred for laser display. Actually, single mode has been reported before in as-prepared spherical caps (*Sci. Rep.* **2**, 244 (2012)). In our work, the BSA is robust enough to support single-mode lasing mode. As shown in Figure R9, single-mode lasing

was achieved when the base diameter of the spherical cap was reduced to 13.5 μm . In addition, the single-mode operation is relatively stable over a large range of pump intensities.

Question 5: *Using an expensive femto-second laser may defeat the purpose for array display. It is possible due to the weak absorption of Red and green dye at UV wavelength (Supplementary Figure 8). Is it possible to pump the micro-laser pixel electrically?*

Response 5: Thanks a lot for the comment. We agree with the referee that using an expensive femto-second (fs) laser as pump source is not appropriate for practical applications of laser display, and realizing electric driven devices is necessary. However, electrically pumped organic lasers still remain a grand challenge and would mark a technological advance with remarkable application potential. So far, a femtosecond laser is still necessary, because its high peak power is essential to amplified spontaneous emission from organic materials. Here, we achieved lasing in all the dye doped spherical caps with an identical fs beam, and the relatively higher lasing thresholds of the green dye than others should be ascribed to the weak absorption at UV band.

In addition, the main stumbling blocks behind electrically pumped organic lasers are the unbalanced carrier mobility and extra triplet losses under electrical injection. It is promising to achieve electrically driven microlaser pixel array based on rational device structures and high-performance optoelectrical materials, such as hybrid perovskites.

Question 6: *It seems the formation of the WGM resonance is the necessary requirement to support the lasing mode at individual pixel. How does that affect the angular distribution of the corresponding stimulated emission modes?*

Response 6: Thanks for the question. Optical cavities are a major component of lasers, surrounding the gain medium and providing feedback of the laser light. Benefiting from the smooth surface and circular base plane, the printed spherical caps function as WGM resonators. We numerically simulated three-dimensional local electric field distribution within the spherical cap with finite-difference time-domain method, which was displayed in Supplementary Figure 11. Notably, the electric field of light is almost localized within the spherical cap, indicating the photons are confined by the total reflection at the interface between the spherical cap and air. The

electric field in the horizontal plane shows a distribution of periodic patterns in the azimuth direction, demonstrating a characteristic WGM resonance. Thus, the angular distribution of the lasing modes is axial symmetric.

Figure R10 Digital images from different positions of an “ICCAS” pattern comprising green emissive spherical caps.

In order to testify the dependence of displays on the angular distribution of lasing modes, we fabricated an “ICCAS” pattern comprising green emissive spherical caps and obtained the digital images with a cellphone from different positions (Figure R10). All the five pictures show clear green ICCAS patterns, demonstrating a broad view-angle of the printed display panel.

Responses to Reviewer 2

Comment: *The manuscript submitted by YS Zhao et al. reports the realization of a display scheme based on a multicolour organic laser array. Each pixel in the array is composed of three individual hemisphere WGM lasers with RGB primary colours, therefore both the colour and the brightness of each pixel can be precisely tuned. The advantage of their approach lies in the simple fabrication in which the laser location and hemisphere droplet size of the lasers can be well determined. Meanwhile, the laser cavity is naturally formed by the hemisphere configuration, thus no external cavity is needed. The related research is very significant and the results reported here are interesting. However, as implementation of laser display is a big engineering issue, the reported demonstration is just an early step towards practical application. Nevertheless, the authors have indeed shown innovation in concept and mechanism in the field of display. In my view, the manuscript may be accepted, subject to the proper addressing of the following comments.*

Response: We thank the referee for the nice summary and very positive evaluation of our work. We are also very grateful for the insightful comments and constructive suggestions, which have helped us to make a further improvement of our work. In the following, we provide concrete responses to the comments and suggestions point-by-point.

Question 1: *The authors have demonstrated both static and dynamic displays, but the obtained images and videos were obtained by a CCD camera. Therefore, the final display performance is still limited by the CCD camera. Then what's the advantage of this approach?*

Response 1: We are very grateful for the reviewer's suggestion. In our design, dynamic displays on the printed RGB microlaser panels were performed by controllably lighting up pixels in different locations through fast scanning of the excitation beam along defined paths on the RGB pixel array. Taking advantage of the human eye's persistence of vision, we were able to observe a complete color image with our eyes when the scanning assignment was fulfilled within the permissible time. Therefore, the display performance is related with pixel panel and scanning laser source, which is not affected by camera. Besides commercial CCD cameras, cell phones were used to record the lasing images and video. Shown in Figure R11 is a display image of a world map, which was taken with the built-in camera of a cell phone. It is worth mention that the

displayed images (Figure 3c in manuscript, and R11) can also be easily captured by the naked eyes. Thus, the final display performance of these printed panels is not limited by CCD camera. However, these images and videos have to be digitally recorded to demonstrate the results with higher quality and better repeatability. Accordingly, we have revised the manuscript in Page 8 as “*With a built-in digital camera in a cell phone, we obtained the far-field image of the fabricated patterns, which can be easily captured by naked eyes. It is shown in Fig. 3c that blue, green and red “ICCAS” patterns appeared when the arrays were fabricated solely with inks doped with S420, uranin and RhB, respectively*”.

Figure R11 Digital image of a word map taken with a cell phone.

Question 2: *As pointed out in this manuscript, for laser display single mode lasers are preferred. Actually, a single mode hemisphere laser has been reported before [Sci. Rep. 2, 244(2012). However, the demonstrated images were obtained by multimode lasers, which actually missed the advantage of laser display. I wonder whether single mode laser display is possible in this approach, considering the individual pixel excitation.*

Response 2: Thanks a lot for the insightful comment. For laser display, single-mode lasers are considered to be the optimal source than multimode lasers due to low noise and good monochromaticity. In comparison with traditional display technologies based on incoherent broadband light sources, laser displays based on multimode lasers still exhibit many advantages, such as wide achievable color gamut, high contrast ratio, and vivid colors. As shown in Figure 3b, we obtained a wide achievable color gamut 45% larger than the standard RGB space by using the multimode RGB laser sources (Fig. 2a-c).

Single mode lasing has been reported in as-prepared spherical caps before (According the suggestion from the other reviewer, we define these printed structures as spherical caps in the revised manuscript for rigorousness). We have studied the lasing characteristics of the individual red, blue and green emissive spherical caps of different size (Supplementary Figure 14), which indicates that the mode numbers decrease with the size reduction. For spherical caps with the same base radius, the red emissive one possesses the least modes among the three types because of the longest lasing wavelength. As shown in Figure R12, single-mode lasing was achieved when the base diameter of the red-emissive spherical cap was reduced to 13.5 μm , and the single-mode operation is relatively stable over a large range of pump intensities. Generally speaking, the cavity loss increases as the size decreases, and thus the lasing becomes more difficult for cavities with smaller cavity length. At present, we cannot achieve single mode lasing from blue and green emissive spherical caps because of the relatively low gain of the dyes, so the demonstrated images and videos were obtained by multimode lasers. We are confident that laser displays based on single-mode lasing can be realized with the inkjet print method by further optimizing the laser dyes.

Figure R12 Photoluminescence spectra from an individual red emissive spherical cap under different pump fluences. Inset shows the corresponding microscopy image. The scale bar is 10 μm .

Question 3: *The final target of laser display will be on either flat panel display or projector display. For projector display, the time duration of lasing should be considered. As current*

approach is using a fs laser for excitation, I wonder how long is the time duration. If the lasing duration is too short, then a CW excitation is necessary.

Response 3: Thanks a lot for the reviewer's helpful comments and suggestions. We agree with the reviewer that time duration of lasing on the pixel has a significant influence on the display effects. The laser image was obtained by consecutively scanning the excitation beam along a corresponding path to light up pixels at specific locations. Taking advantage of the human eye's persistence of vision, we were able to observe a complete color image when the scanning assignment was fulfilled within the permissible time ($\sim 1/50$ second). So the time duration of lasing on a pixel depending on the laser scanning speed, is on the order of millisecond. This time is long enough for fs laser to excite the pixel to achieve lasing, and the laser signal can also be captured by human eyes.

On the other hand, using an expensive femto-second (fs) laser as pump source is not appropriate for practical applications of laser display, and realizing electric driven devices is necessary. However, electrically pumped organic lasers still remain a grand challenge and would mark a technological advance with remarkable application potential. Thus, CW pumped lasing would be a better choice for laser displays due to the low cost and easy integration.

Question 4: *For flat panel display, electrical driven devices are necessary, Can the authors comment on the feasibility of electrical pumping?*

Response 4: We totally agree with the reviewer that the electrical driven devices are necessary. Using an expensive femto-second (fs) laser as pump source is not appropriate for practical applications of laser display. However, electrically pumped organic lasers still remain a grand challenge in this field and would mark a big technological advance with remarkable application potential. So far, a fs laser is still necessary, because its high peak power is essential to amplified spontaneous emission from organic materials. Here, we achieved lasing in all the spherical caps doped with different dyes under the pump of an identical fs laser beam, and the relatively higher lasing thresholds of the green dye can be ascribed to the weak absorption at UV band.

In addition, the main stumbling blocks behind electrically pumped organic lasers are the unbalanced carrier mobility and extra triplet losses under electrical injection. It should be promising to achieve electrically driven microlaser pixel array by designing optoelectrical materials with higher combination properties. Organic semiconductor materials with both high

charge carrier mobility and high luminescence efficiency in the solid state would be beneficial. It is also very important to avoid the Auger recombination and triplet annihilation. At present, thermally activated delayed fluorescence (TADF) dyes should be a promising type of materials for achieving the electrically driven lasing by effectively harvesting the triplet excitons, In addition, some luminophores with aggregation-induced emission enhancement (AIEE) should be another solution to this problem because the luminescence quantum yield and charge carrier mobility can be simultaneously increased at crystal state.

REVIEWERS' COMMENTS:

Reviewer #1 (Remarks to the Author):

Authors have thoroughly revised manuscript and provided substantial new results in this revision. I believe my previous questions have been sufficiently addressed. I would like to recommend this manuscript to be accepted for publication.

Cheng Sun

Reviewer #2 (Remarks to the Author):

I have carefully read the revised version and the Response to Reviewers' Comments. My opinion is that all the comments have been well answered by the authors and the revised paper meets the publication criteria. Laser based display possesses clear advantages in terms of display quality and energy efficiency. The work reported in this paper seems to be conceptually novel and technically sound. It may attract broad attention. Although it is still far from the practical application, the work has paved a way. I recommend for publication in Nature Communications.

Handong Sun